# Low knowledge about hepatitis B prevention among pregnant women in Kinshasa, Democratic Republic of Congo

Sahal Thahir[1‡], Samantha E. Tulenko[2‡*], Patrick Ngimbi[3], Sarah Ntambua[3], Jolie Matondo[3], Kashamuka Mwandagalirwa[3], Martine Tabala[3], Didine Kaba[3], Marcel Yotebieng[4], Jonathan B. Parr[5], Peyton Thompson[1]

1 Division of Infectious Diseases, Department of Pediatrics, University of North Carolina at Chapel Hill, Chapel Hill, North Carolina, United States of America, 2 Department of Epidemiology, UNC Gillings School of Global Public Health, Chapel Hill, North Carolina, United States of America, 3 Kinshasa School of Public Health, Kinshasa, Democratic Republic of the Congo, 4 Division of General Internal Medicine, Department of Medicine, Albert Einstein College of Medicine, New York, New York, United States of America, 5 Division of Infectious Diseases, Department of Medicine, University of North Carolina at Chapel Hill, Chapel Hill, North Carolina, United States of America

‡ These authors share first authorship on this work.
* samantha_tulenko@med.unc.edu

**Data Availability Statement:** This study's underlying data has been published through the Carolina Digital Repository and is accessible at

## Abstract

Infants infected perinatally with hepatitis B (HBV) are at the highest risk of developing chronic hepatitis and associated sequelae. Prevention of mother-to-child transmission (PMTCT) of HBV requires improved screening and awareness of the disease. This study evaluated existing HBV knowledge among pregnant mothers (n = 280) enrolled in two HBV studies in urban maternity centers in Kinshasa, Democratic Republic of the Congo. All mothers responded to three knowledge questions upon study enrollment. Baseline levels of knowledge related to HBV transmission, treatment, prevention, and symptoms were low across all participants: 68.8% did not know how HBV was transmitted, 70.7% did not know how to prevent or treat HBV MTCT, and 79.6% did not know signs and symptoms of HBV. Over half of participants responded "I don't know" to all questions. HBV-positive women who participated in both studies (n = 46) were asked the same questions during both studies and showed improved knowledge after screening and treatment, despite no formal educational component in either study (p < 0.001). These findings highlight the need for intensified education initiatives in highly endemic areas to improve PMTCT efforts.

## Introduction

Hepatitis B virus (HBV) is one of the most common causes of chronic hepatitis in the world, with highest prevalence in sub-Saharan Africa (SSA) and the Western Pacific region.[1] Despite an effective vaccine, an estimated 1.5 million new infections occurred globally in 2019; almost 1 million of these new infections occurred in Africa [1]. Most chronic infections in SSA result from HBV transmission that occurs before adolescence, either from mother-to-child

https://cdr.lib.unc.edu/concern/data_sets/9306t874x.

**Funding:** The AVERT study was funded by the Gillings Innovation Laboratory award – funded by the 2007 Gillings Gift to the University of North Carolina–Chapel Hill's Gillings School of Global Public Health. The BDI study was funded by a postdoctoral fellowship in tropical medicine from ASTMH/Burroughs-Wellcome Fund granted to PT. The HIV prevention of mother-to-child transmission study whose infrastructure was leveraged for this project was supported by NIH grants (NICHD R01HD087993 and NIAID U01AI096299; PI Yotebieng). SET is funded by an NIH pre-doctoral training grant (F30AI164818). PT received salary support from an NIH grant (NIAID K08AI148607). The funders had no role in study design, data collection and analysis, decision to publish, or preparation of the manuscript.

**Competing interests:** The authors have read the journal's policy and have the following competing interests: Outside of the submitted work, JBP, PT, and MY report research support from Gilead Sciences and non-financial support from Abbott Diagnostics; JBP has also received research support from the World Health Organization and an honorarium from Virology Education. This does not alter our adherence to PLOS policies on sharing data and materials.

transmission (MTCT) or horizontal transmission among children [2]. MTCT is responsible for one third of chronic HBV infections worldwide, and an estimated 15–40% of persons chronically infected will develop HBV-related sequelae, such as cirrhosis and hepatic carcinoma [3]. Furthermore, MTCT is a major factor leading to HBV endemicity, as 70–90% of mothers with high-risk HBV infection (viral load $\geq$ 200,000 IU/mL or HBeAg positivity) will transmit the virus to their infants in the absence of any intervention [4]. Even among mothers without high-risk HBV infections, the rate of transmission is 10–40% [4].

Therefore, prevention of MTCT (PMTCT) is critical to addressing the HBV epidemic [5]. The WHO recommends a three-dose HBV vaccine series beginning within 24 hours of delivery; this schedule has been shown to protect over 95% of infants against HBV infection [6,7]. However, breakthrough transmission can occur in infants born to women with high-risk HBV infection despite vaccination [7,8]. In 2020, the WHO released guidelines on HBV PMTCT recommending that women with high-risk HBV infection receive antiviral prophylaxis, starting from the 28[th] week of pregnancy [6]. Effective HBV PMTCT strategies in endemic regions require appropriate funding and infrastructure as well effective participation of pregnant women in screening and treatment through pregnancy, delivery, and the post-partum period. To date, there are limited studies which evaluate the perceptions and knowledge of HBV among pregnant women in endemic regions [7,9–11].

Chronic HBV is highly endemic in the Democratic Republic of the Congo (DRC). HBV prevalence is approximately 3–5%, and approximately 300,000 Congolese children younger than 5 years live with chronic HBV infection [12]. Currently, the DRC is evaluating potential HBV screening, birth-dose vaccination, and treatment programs for pregnant women in the DRC. Here, we assess HBV knowledge and perceptions among pregnant mothers in urban Kinshasa, DRC in preparation for future implementation of a national hepatitis elimination plan.

## Methods

In this study, we performed a secondary analysis of data collected during two studies of HBV prevention of maternal-to-child transmission (PMTCT) in Kinshasa, DRC: The Arresting Vertical Transmission of HBV (AVERT) study [12] and the ongoing Birth Dose Immunogenicity (BDI) study (**Fig 1**). Both studies were conducted in two large, private, not-for-profit facilities:

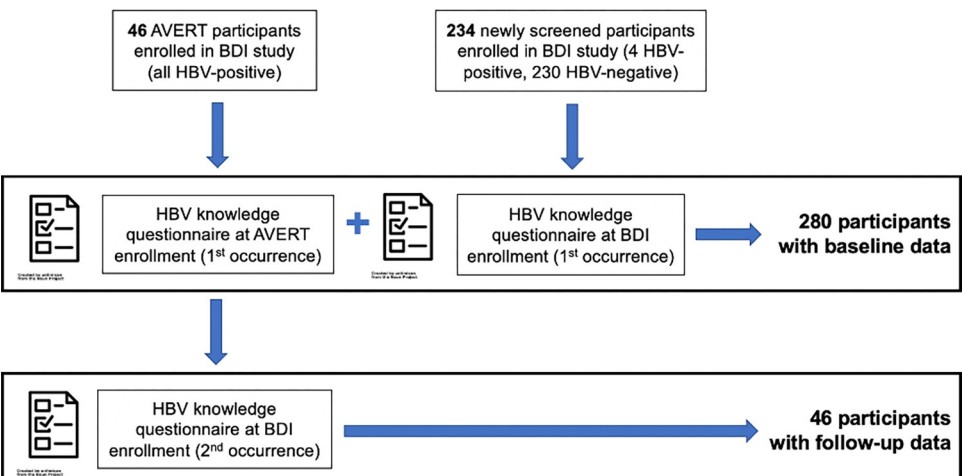

**Fig 1. Schematic depicting activities in the AVERT and BDI studies, including the point at which enrollment questionnaires were conducted.**

Kingasani and Binza maternity centers. The AVERT study was developed to determine the feasibility of HBV PMTCT through testing and treating pregnant women and providing birth-dose vaccination for exposed infants. The aim of the BDI study (clinicaltrials.gov NCT03897946) was to evaluate the immunogenicity of an added HBV birth-dose vaccination to the infant immunization schedule in HBV-exposed and HBV-unexposed infants. Mothers in the AVERT study were HBV-positive women who enrolled during prenatal visits in 2018–2019. Mothers in the BDI study included previously identified HBV-positive women from AVERT who consented to join BDI, as well as newly screened, mostly HBV-negative women who were enrolled when they presented for delivery in 2019 (**Fig 1**). Upon initial study enrollment, participants received group counseling on HBV and PMTCT and underwent point-of-care HBV surface antigen (HBsAg) testing. Participants also completed a standardized questionnaire on demographics and clinical characteristics; this questionnaire included a section on HBV knowledge regarding modes of transmission, signs and symptoms of infection, PMTCT measures, and perceived severity of HBV infection (**S1 Questionnaire**). Mothers who participated in both AVERT and BDI completed the questionnaire at AVERT enrollment and again approximately 9 months later at BDI enrollment.

Institutional review board approval was obtained from the University of North Carolina (17–2090 [AVERT] and 18–2793 [BDI]) and from Kinshasa School of Public Health (ESP/CE/021/2019 and ESP/CE/062/2019, respectively). Both studies were registered at clinicaltrials.gov (NCT03567382 and NCT03897946, respectively). Written informed consent was obtained from all subjects.

In the present study, we used mothers' responses to the baseline questionnaires collected during the AVERT and BDI studies to assess HBV knowledge among participants in both studies. HBV knowledge scores were calculated as the sum of up to 9 correct answers to 3 knowledge questions asked in the questionnaire (**S1 Questionnaire**). Participants could choose multiple answers to each question. A score of 1 corresponds with 1 correct answer in response to a knowledge question and a score of 9 corresponds to correct answers to all 9 possible answer choices. For analysis purposes, we dichotomized HBV knowledge as "any knowledge" (score on knowledge assessment > 0) or "no knowledge" (score on knowledge assessment = 0). All study participants completed at least one baseline questionnaire. We assessed baseline HBV knowledge from each participant's first questionnaire and demographic characteristics using descriptive statistics. The subset of women enrolled in both the AVERT and BDI studies completed the questionnaire twice and were assessed for both baseline HBV knowledge and change in HBV knowledge following participation in the AVERT study (**Fig 1**). Among AVERT mothers, we compared pre- and post-enrollment HBV knowledge scores using a paired t-test. All analyses were conducted using R (Version 1.4.1717).

## Results

### Study population

The study population consisted of 280 pregnant women recruited from Binza and Kingasani maternity centers in Kinshasa Province as part of the AVERT and BDI trials. Fifty women (17.9%) were HBV-positive and 230 (82.1%) were HBV-negative. Of the 50 HBV-positive women, 46 (92.0%) were enrolled in the AVERT trial prior to participating in the BDI study and 4 (8.0%) screened positive during BDI enrollment. All 280 women responded to each of the three knowledge questions; there was no missing data.

### Baseline knowledge assessment

On the baseline knowledge assessment, the average HBV knowledge score among all women was 1.08. Scores ranged from 0 to 6 out of 9; no participants answered all questions correctly.

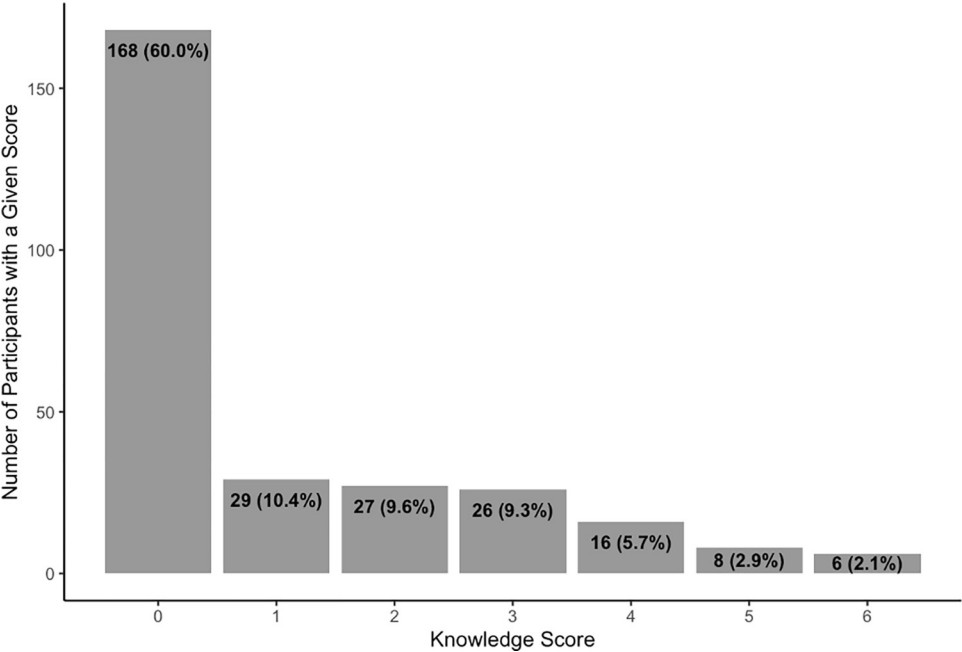

**Fig 2. Distribution of HBV knowledge scores at baseline.** Score reflects the number of correct responses to 9 statements about HBV symptoms, transmission, and prevention. The average knowledge score was 1.06.

The majority of women (60.0%) had no correct answers, receiving a score of 0 on our 9-point HBV knowledge assessment scale (**Fig 2**). All of these women responded "I don't know" in response to each HBV question. Women with no HBV knowledge were slightly younger than women with any HBV knowledge (mean age 28.1 vs 29.5 years), More women with no HBV knowledge were HBV-negative, attended Binza maternity center, and had worked in the last 7 days (**Table 1**). Nearly all (97.5%) of the women had attended secondary school or a higher education level; education level did not differ by HBV knowledge. Within each section of the questionnaire, most women responded that they did not know the correct answer: 233 women (79.6%) did not know the signs and symptoms of HBV, 192 women (68.8%) did not know how a person could acquire HBV, and 198 women (70.7%) did not know how MTCT of HBV could be prevented (**Table 2**).

## Perception of HBV

In response to the question, "In your opinion, how serious a disease is Hepatitis B?", most women (70.7%) selected "very serious", and 20.7% selected "I don't know" (**Fig 3**). A greater proportion of women with any HBV knowledge considered HBV "very serious" compared with women with no HBV knowledge (77.7% vs. 66.1%). More women with no HBV knowledge responded that they "didn't know" about HBV severity (28.6% vs. 8.9%).

## Follow-up HBV knowledge assessment

Among the 46 women who participated in both studies, the average knowledge score at baseline was 2.8 (range 0–6) and average score at follow-up was 3.7 (range 0–7) (p < 0.001) (**Fig 4**). Three answer choices in particular had a higher proportion of correct responses at follow-up compared to baseline: (1) recognizing jaundice as a symptom of HBV (52.2% correctly reported at baseline vs. 91.3% at follow-up), (2) recognizing that a person can contract HBV

**Table 1. Demographic characteristics of enrolled women in the overall cohort and stratified by HBV knowledge category.** Participants in the "any HBV knowledge" category scored > 0 on the HBV knowledge assessment and participants in the "no HBV knowledge" category scored 0 points on the HBV knowledge assessment.

| Characteristic | Overall (n = 280) | Any HBV Knowledge (n = 112) | No HBV Knowledge (n = 168) |
|---|---|---|---|
| Mean age in years (range) | 28.7 (18–45) | 29.5 (18–42) | 28.1 (18–45) |
| Maternal HBV status | | | |
| HBV-positive | 50 (17.9%) | 38 (33.9%) | 12 (7.1%) |
| HBV-negative | 230 (82.1%) | 74 (66.1%) | 156 (92.9%) |
| Maternity center attended | | | |
| Binza | 143 (51.1%) | 17 (15.2%) | 126 (75.0%) |
| Kingasani | 137 (48.9%) | 95 (84.8%) | 42 (25.0%) |
| Gravidity | | | |
| 1–2 pregnancies | 149 (53.2%) | 53 (47.3%) | 96 (57.1%) |
| > 2 pregnancies | 131 (46.8%) | 59 (52.7%) | 72 (42.9%) |
| Marital status | | | |
| Married | 222 (79.3%) | 86 (76.8%) | 136 (81.0%) |
| Unmarried | 58 (20.7%) | 26 (23.2%) | 32 (19.0%) |
| Education | | | |
| No school | 2 (0.7%) | 2 (1.8%) | 0 |
| Primary school | 5 (1.8%) | 2 (1.8%) | 3 (1.8%) |
| Secondary school | 200 (71.4%) | 80 (71.4%) | 120 (71.4%) |
| Higher education | 73 (26.1%) | 28 (25.0%) | 45 (26.8%) |
| Work in the last 7 days | | | |
| No work | 175 (62.5%) | 60 (53.6%) | 115 (68.5%) |
| Any work | 105 (37.5%) | 52 (46.4%) | 53 (31.5%) |
| Wealth quartile | | | |
| Lowest quartile | 70 (25.0%) | 24 (21.4%) | 46 (27.4%) |
| Lower-to-middle | 70 (25.0%) | 22 (19.6%) | 48 (28.6%) |
| Upper-to-middle | 70 (25.0%) | 32 (28.6%) | 38 (22.6%) |
| Highest quartile | 70 (25.0%) | 34 (30.4%) | 36 (21.4%) |

through contact with blood from an infected person (54.3% vs. 89.1%), and (3) recognizing that MTCT of HBV can be prevented by infant vaccination at birth (15.2% vs. 76.1%). At follow-up, fewer women responded "I don't know" to all questions (8 women at baseline [17.4%] vs. 1 woman at follow-up [2.2%]).

## Discussion

In this analysis of data collected during two HBV PMTCT studies in Kinshasa, DRC, HBV knowledge was low among pregnant women with regards to HBV transmission, symptoms, treatment, and prevention of MTCT. Nearly two-thirds of women responded "I don't know" to all questions asked. HBV-positive mothers enrolled in the AVERT study displayed significant improvement in HBV knowledge through study participation, despite the lack of formal education in the study design.

This observed lack of HBV knowledge among 60% of participants is consistent with previous studies of pregnant populations in sub-Saharan Africa, in which 50% to 76% of participants had low HBV knowledge [10,13–17]. In a recent study from a rural area of DRC, authors found that most pregnant women presenting for care lacked HBV knowledge: 93% of study participants had no knowledge of HBV [18]. Mudji et al. attributed this low knowledge in part

**Table 2. Distribution of responses to the three questions assessing Hepatitis B knowledge among all enrolled women.** 280 total participants responded to each question. Because participants could choose multiple answers to each question, percent of participants who chose each answer sums to greater than 100% for each question. The percent of participants who chose each answer was calculated using total number of participants as the denominator, not total number of answers chosen.

**Q1: What are the signs and symptoms of Hepatitis B Virus?**

| Select all that apply: | Number (%) of participants who chose this answer |
|---|---|
| No symptoms | 3 (1.1%) |
| Jaundice (yellowing of skin) | 42 (15.0%) |
| Abdominal pain | 18 (6.4%) |
| Rash | 9 (3.2%) |
| Don't know | 223 (79.6%) |
| Refuse to answer | 0 (0%) |

**Q2: How can a person get HBV?**

| Select all that apply: | Number (%) of participants who chose this answer |
|---|---|
| Handshakes | 2 (0.7%) |
| Contact with infected person's blood | 63 (22.5%) |
| Sharing dishes | 1 (0.4%) |
| Mother-to-child transmission | 25 (8.9%) |
| Sexual transmission | 48 (17.1%) |
| Breastfeeding | 0 (0%) |
| Don't know | 192 (68.6%) |
| Refuse to answer | 1 (0.4%) |

**Q3: How can mother-to-child transmission of HBV be prevented?**

| Select all that apply: | Number (%) of participants who chose this answer |
|---|---|
| Infant vaccination at birth | 42 (15.0%) |
| Antiviral treatment during pregnancy | 51 (18.2%) |
| Don't know | 198 (70.7%) |
| Refuse to answer | 1 (0.4%) |

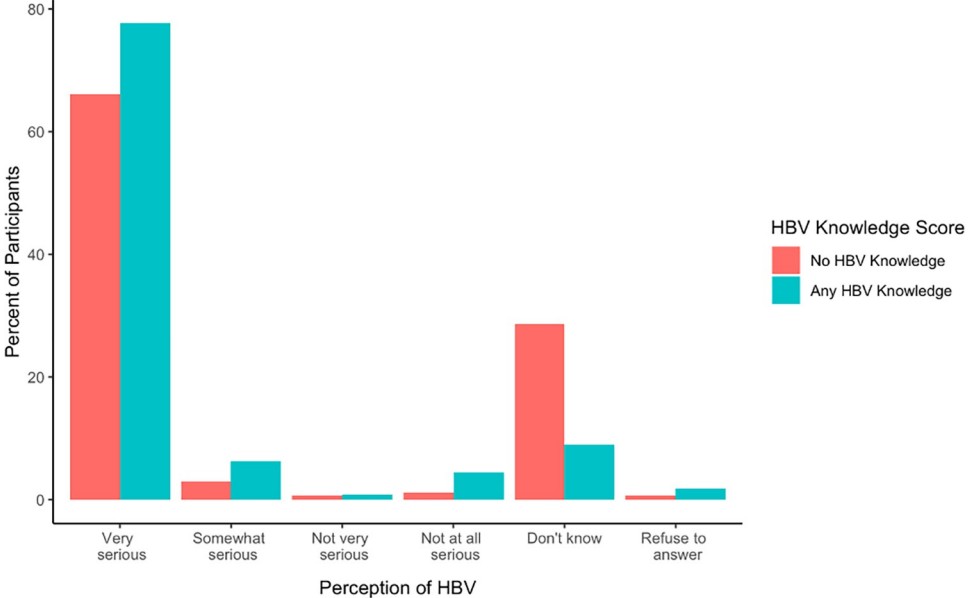

**Fig 3. Participants responses to the question "in your opinion, how serious a disease is hepatitis B?" among women with no HBV knowledge (score of 0 on HBV knowledge assessment) and women with any HBV knowledge (score > 0 on HBV knowledge assessment).**

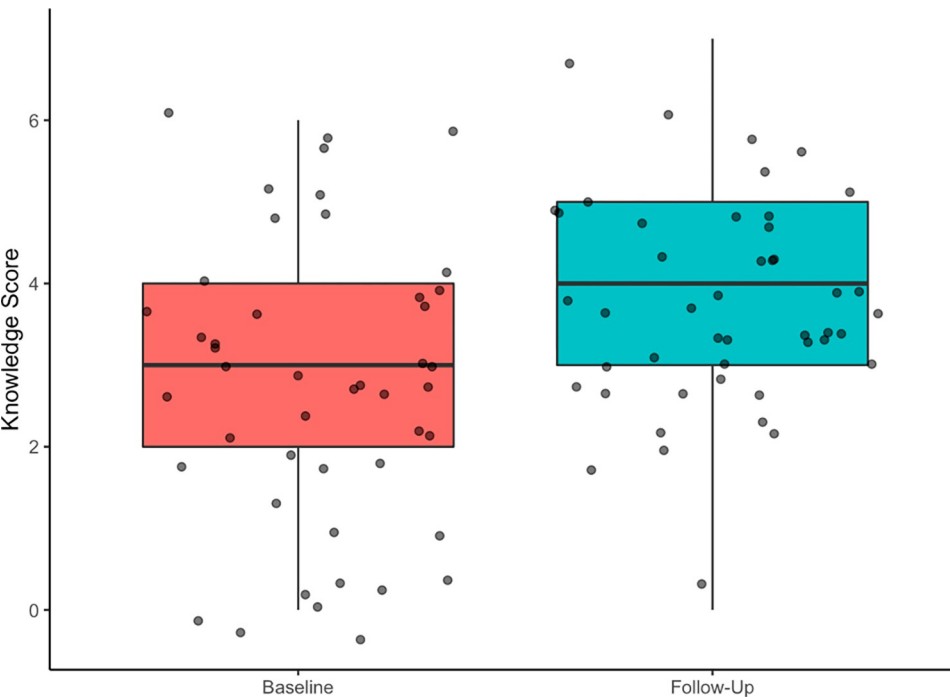

**Fig 4. Distribution of knowledge scores at baseline and follow-up among 46 HBV-positive women who participated in both studies.** Average score at baseline was 2.8 (range 0–6) and average score at follow-up was 3.7 (range 0–7) (p < 0.001).

to less health education in more rural areas. In contrast, our study took place in urban Kinshasa. Additional studies in Cameroon, Ethiopia, Kenya, and Ghana found that higher levels of HBV knowledge were associated with increased level of education attained [10,15,16,19]. In particular, Gebrechokos et al. noted that factors such as education level and higher monthly income were associated with adequate HBV knowledge, likely due to improved access to media and technologies that allowed for access to HBV-related information [10]. However, in our study, low education did not appear to explain lack of HBV knowledge, as 97.5% of participants had attended at least secondary school (**Table 1**). Instead, low HBV knowledge might be due to a lack of public education regarding HBV in the DRC, highlighting an important gap in public health messaging.

Interestingly, we did identify some characteristics with differential distributions between participants with any HBV knowledge and those with no HBV knowledge (**Table 1**). These observed associations should be interpreted cautiously given that this study was not designed to assess causal relationships between demographic factors and HBV knowledge. For example, the mean participant age was greater among participants with any HBV knowledge; however, this likely reflects the fact that HBV-positive participants were older and HBV-positive participants were more likely to have any HBV knowledge. Strikingly, a substantially higher proportion of participants from Kingasani maternity had any HBV knowledge compared with participants from Binza. Both facilities are private, Catholic maternity centers located in Kinshasa, but their attendants differ sociodemographically. It is possible the observed difference in knowledge at the two facilities underscores the influence of other, unmeasured sociodemographic differences on HBV knowledge. Futhermore, the research and educational practices of the two facilities may have influenced study results, as Kingsani maternity has historically been involved in more clinical research studies. More participants with no HBV knowledge were in the lower and low-to-middle quartiles for wealth compared with participants with any HBV

knowledge. Also, more participants with no HBV knowledge did not report working in the past 7 days (**Table 1**). These are pertinent associations to consider in future, larger studies about HBV knowledge among pregnant women.

Encouragingly, we observed improvement in HBV knowledge over time among HBV-positive mothers who participated in the AVERT trial.[13] These women received brief, informal HBV education upon enrollment, and participation in study activities might also have passively increased awareness of HBV. Alternatively, this knowledge increase may have been due to frequent interactions with healthcare workers during the treatment period, which has been shown to improve communication and trust in healthcare workers [20–22]. Participation in the study may have increased the confidence of these women and their comfort in interacting with study staff and in responding to the questionnaire. Of note, HBV-positive women had higher average knowledge scores than HBV-negative women at baseline, suggesting these women may have differed at baseline. Further research is needed to understand the mechanism for this observed increase in knowledge to inform future educational interventions.

While these findings provide important insight into HBV knowledge levels among pregnant women in Kinshasa, this study is limited by its retrospective design and secondary data analysis from study questionnaires that were not designed to comprehensively assess HBV knowledge and perceptions. Additionally, the observed improvement in scores among participants in both the AVERT and BDI studies could be influenced by selection bias. It is possible that participants who agreed to participate in both studies were inherently more likely to engage in knowledge-seeking behaviors after learning of their HBV diagnosis than the average person. Thus, we cannot definitively say whether the participants with baseline and follow-up knowledge scores are representative of our whole population of interest. Finally, study participants were recruited from large maternity centers in urban Kinshasa, and findings may not be generalizable to other areas of the DRC where access to healthcare and public health education is generally more limited. However, given the low knowledge scores among this relatively well-educated cohort, it is reasonable to infer that HBV knowledge may be even lower in other settings in the DRC.

## Conclusion

The low baseline HBV knowledge scores observed in this study suggest that HBV education efforts for pregnant women should be prioritized as part of HBV elimination efforts. The improvement in HBV knowledge observed over the course of the AVERT study suggests that one strategy might be to integrate HBV testing and treatment into routine prenatal care alongside a rigorous educational initiative. Future studies that pilot HBV educational interventions for pregnant women are urgently needed in sub-Saharan Africa.

## Supporting information

**S1 Questionnaire. Enrollment questionnaire assessing knowledge, attitudes, and beliefs of pregnant women in the DRC.** This questionnaire was used upon enrollment of both the AVERT and BDI trials. Denoted "incorrect" or "correct" in *italics* for the purposes of this manuscript (not included in the original knowledge questionnaire).
(DOCX)

## Acknowledgments

We thank all of the women who participated in this study, the staff at the Binza and Kingasani maternity clinics, and provincial and national health authorities. We would like to thank Dr.

Noro Ravelomanana, Dr. Malongo Fathy, and Dr. Bienvenu Kawende for their contributions to this study.

We are grateful for the support from the administrative staff at the Kinshasa School of Public Health and at the University of North Carolina. We grieve the loss of Dr. Steven Meshnick, whose vision and mentorship were critical to the success of this study.

## Author Contributions

**Conceptualization:** Patrick Ngimbi, Sarah Ntambua, Jolie Matondo, Kashamuka Mwandagalirwa, Martine Tabala, Didine Kaba, Marcel Yotebieng, Jonathan B. Parr, Peyton Thompson.

**Data curation:** Sahal Thahir, Samantha E. Tulenko, Patrick Ngimbi.

**Formal analysis:** Sahal Thahir, Samantha E. Tulenko.

**Funding acquisition:** Marcel Yotebieng, Peyton Thompson.

**Methodology:** Sahal Thahir, Samantha E. Tulenko.

**Project administration:** Patrick Ngimbi, Sarah Ntambua, Jolie Matondo, Kashamuka Mwandagalirwa, Martine Tabala.

**Visualization:** Samantha E. Tulenko.

**Writing – original draft:** Sahal Thahir, Samantha E. Tulenko.

**Writing – review & editing:** Sahal Thahir, Samantha E. Tulenko, Patrick Ngimbi, Sarah Ntambua, Jolie Matondo, Kashamuka Mwandagalirwa, Martine Tabala, Didine Kaba, Marcel Yotebieng, Jonathan B. Parr, Peyton Thompson.

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
