## [Decision Letter · Decision Letter 0]

9 May 2022

PGPH-D-22-00597

Low knowledge about hepatitis B prevention among pregnant women in the Democratic Republic of the Congo

Dear Dr. Tulenko,

Thank you for submitting your manuscript to PLOS Global Public Health. After careful consideration, we feel that it has merit but does not fully meet PLOS Global Public Health’s publication criteria as it currently stands. Therefore, we invite you to submit a revised version of the manuscript that addresses the points raised during the review process.

Please submit your revised manuscript by . If you will need more time than this to complete your revisions, please reply to this message or contact the journal office at globalpubhealth@plos.org. Please include the following items when submitting your revised manuscript:

We look forward to receiving your revised manuscript.

Kind regards,

Karen D. Cowgill, PhD, MSc

Academic Editor

Journal Requirements:

2. We noticed that you used “data not shown”/"unpublished data" in the manuscript. We do not allow these references, as the PLOS data access policy requires that all data be either published with the manuscript or made available in a publicly accessible database. Please amend the supplementary material to include the referenced data or remove the references.

3. Your manuscript is missing the following sections: Introduction. Please ensure these are present, and in the correct order, and that any references to subheadings in your main text are correct. An outline of the required sections can be consulted in our submission guidelines here:

https://journals.plos.org/globalpublichealth/s/submission-guidelines#loc-parts-of-a-submission

4. We have noticed that you have uploaded Supporting Information files, but you have not included a list of legends. Please add a full list of legends for your Supporting Information files after the references list. 

Additional Editor Comments (if provided):

Reviewers' comments:

Reviewer's Responses to Questions

**Comments to the Author**

1. Does this manuscript meet PLOS Global Public Health’s publication criteria? Is the manuscript technically sound, and do the data support the conclusions? The manuscript must describe methodologically and ethically rigorous research with conclusions that are appropriately drawn based on the data presented.

Reviewer #1: Partly

Reviewer #2: Yes

Reviewer #3: Yes

2. Has the statistical analysis been performed appropriately and rigorously?

Reviewer #1: No

Reviewer #2: I don't know

Reviewer #3: Yes

3. Have the authors made all data underlying the findings in their manuscript fully available (please refer to the Data Availability Statement at the start of the manuscript PDF file)?

Reviewer #1: Yes

Reviewer #2: Yes

Reviewer #3: Yes

4. Is the manuscript presented in an intelligible fashion and written in standard English?

Reviewer #1: Yes

Reviewer #2: Yes

Reviewer #3: Yes

5. Review Comments to the Author

Reviewer #1: Dear authors,

I enjoyed reading your paper on this very important matter in the DRC. Reading the results on the low awareness and knowledge about HBV shows how important this subject matter is for public health prevention strategies. Thank you for addressing this issue and putting it in numbers.

I find the manuscript as it is presented now not suitable for publication. Because of the high importance of the topic, I suggest a major revision of the manuscript. The manuscript has room for textual improvement, more scientific writing, improved structure and flow, use of subheadings. For feedback on each part of the manuscript please see below. I hope this review will help you to improve the manuscript.

Introduction

I find the introduction a bit short where more information could be provided about the HBV status in the DRC, insights on the numbers in DRC, why you want to target this specific group, a problem statement. L45, do you have estimates on the high endemic HBV status of the DRC?

Methods

The method section states that a secondary analysis was conducted. This means that you did not have to collect the data again but rather used the existing data. If this is so, why do you report on the written informed consent and the ethical approval? It was to me, as a peer-reviewer, not very clear on how you did the secondary analysis, what part of the BDI and AVERT study data you used. Could you please rewrite this section and clarify? In L81 you describe a sum of scores, however the total sum, its meaning, cut off values etc are not described. The reader has to know how to interpret the values e.g. is 9 high or low?

Results

Could you please restructure the results, use subheadings and also conduct statistical analysis on factors that might be associated with HBV knowledge.

L93- L96: this information belongs in the methods section.

L103: you introduce this info: 0 on our 9-point HBV knowledge assessment scale. Could you please address this scale in the methods section?

L108: women were slighetly younger. What does this mean? Was age a significant factor? Could you add the data to the text?

L153 – L154: this information should be in the text

Conclusion:

I believe this heading should be changed into discussion. The first paragraph of the discussion section should wrap up the main results of the study prior to discussing these results in the rest of the discussion section. You can add a final paraghraph named conclusion. Also, I am missing what your data really means in the context. I like that you have drafted recommendations.

L162: I think you mean ‘had lower HBV knowledge at baseline’

L169: you write that education was not significantly associated with hbv knowledge. However, I could not retrieve this data in the tables nor in the results section.

I hope to have informed you sufficiently.

Good luck.

Reviewer #2: Comments

The title said “Low knowledge about hepatitis B prevention among pregnant women in the Democratic Republic of the Congo” as my view it could be good to say “To assess the knowledge or to determine the knowledge level…….” Rather than saying Low knowledge. To say a low knowledge, we have to have some measurement to categorize it low or high or other. Another point here in the title that need attention is the scope of the study; it said “ ……among the pregnant women in the DRC” it is better to put the province or county or district of the study rather than generalizing the finding of two health facility to the country level. In clear term the data that the study obtained might not represent the country as whole so it needs some modification.

In the abstract part the study is summarized from line 27 to 33, it is better if the major finding of the knowledge status of each question is putted. In addition, adding the key word at the end of abstract is make ease to the reader to find on search engine, thus it is better to consider keyword of the study.

The intention of introducing of HBV magnitude that used in introduction part is slightly differ from the sentence that used in abstract part line 23 and 24…. Infants infected perinatally with hepatitis B (HBV) are at the highest risk …… so it is also better to more elaborate this sentence in introduction part or boldly show it.

In method part the sample size determination and sampling technique is expected, it is fascinating to mention it.

On other hand in method part the issue of data quality, how the questionnaire or data collection tool was prepared and validated has to stated. But I fail to get it, so it is better to state the brief of it. Another important point here is how the study operationalizes the knowledge measurement. This study has to incorporate and elaborate how some very pertinent variables were operationalized with its reference if any.

Under method part of Line 81 the calculation of knowledge was stated, it is good if the study shows us the cutoff point that show the level of knowledge is mentioned.

The data analysis part is need an important point that determine the statistical analysis is significant or not, meanwhile the p value or other measurement that declare the significance level has to mentioned.

In result part the response rate mentioned on line 118 has to move to the top of result narration paragraph. In addition, it is better to move the statement that mentioned under line 119 and 120 to method part.

Regarding the table of sociodemographic characteristics as to me it is better to make it the first table of the article, and also the table need the top and bottom border.

The statement that mentioned on line 125 and 126 is not clear for me.

Question

1. Did the author believe that the title of study is clearly consistent with result of the study?

2. In result part of table 2, the table grouped as Any HBV Knowledge and No HBV Knowledge. What do the No HBV knowledge mean? It needs more clarification under method part.

3. Did the author convinced that all statistical analysis were performed accordingly?

3. In discussion part the result of study is compared in percent with another study finding that also mentioned in percent. How clearly we know whether the finding is higher than of lower that without showing the exact confidence interval of the percent?

After all the paper is potentially publishable once the authors revised their manuscript based on the comments given and responded to each questions accordingly.

Kind regards.

Takala Utura (MPH)

School of Public Health, Institute of Health, Bule Hora University, Ethiopia

PO. Box 144

Email: takekerrofamily@gmail.com

Mobile: +251912452736

Reviewer #3: I would like to thank you for giving me the opportunity to review this well-written manuscript and congratulate the team on their excellent work. Comments on this manuscript primarily focuses on the analysis and interpretation of the data.

Overall, it would be helpful to include information on the distribution of the responses (with HBV knowledge or without HBV knowledge) by different characteristics of the study population as a whole. For example, at the Maternity Center in Binza, the percentages of women who had HBV knowledge versus those who had no HBV knowledge were 11.9% (17/143) and 88.1% (126/143), respectively, whereas at Kingasani, the percentages were 69.3% (95/137) and 30.7% (42/137) respectively. Thus, these data clearly indicated that there was a significant geographical variation in knowledge of HBV. Ideally, the team should continue to examine and stratify the geographic factors (variations) that may influence or explain other factors such as wealth, work, and education to ensure that no confounding factors might affect the conclusions of this study. In addition, it appears that the work, wealth quartile factor may also be related to knowledge of HBV to some degree, based on the data.

Lastly, regarding the findings of improving knowledge of HBV over the course of the AVERT trial, based on the design of the study, there might be a bias related to "selection bias" in which the participants who agreed to participate in BDI, regardless of the informal education received from the program, had an inherent, unconscious tendency to actively seek knowledge to manage the conditions once they became aware of the conditions. Additional analysis/explanation may be required to distinguish and quantify the efforts between self-contribution factors and external influences from informal education that contribute to the improvement of HVB knowledge.

6. PLOS authors have the option to publish the peer review history of their article (what does this mean?). If published, this will include your full peer review and any attached files.

**Do you want your identity to be public for this peer review?** For information about this choice, including consent withdrawal, please see our Privacy Policy.

Reviewer #1: No

Reviewer #2: No

Reviewer #3: No

---

## [Decision Letter · Decision Letter 1]

19 Aug 2022

Low knowledge about hepatitis B prevention among pregnant women in Kinshasa, Democratic Republic of Congo

PGPH-D-22-00597R1

Dear Samantha E Tulenko,

We are pleased to inform you that your manuscript 'Low knowledge about hepatitis B prevention among pregnant women in Kinshasa, Democratic Republic of Congo' has been provisionally accepted for publication in PLOS Global Public Health.

Best regards,

Edina Amponsah-Dacosta, Ph.D., MPH

Academic Editor

Reviewer Comments (if any, and for reference):

Reviewer's Responses to Questions

**Comments to the Author**

1. If the authors have adequately addressed your comments raised in a previous round of review and you feel that this manuscript is now acceptable for publication, you may indicate that here to bypass the “Comments to the Author” section, enter your conflict of interest statement in the “Confidential to Editor” section, and submit your "Accept" recommendation.

Reviewer #1: All comments have been addressed

Reviewer #2: All comments have been addressed

Reviewer #3: All comments have been addressed

2. Does this manuscript meet PLOS Global Public Health’s publication criteria? Is the manuscript technically sound, and do the data support the conclusions? The manuscript must describe methodologically and ethically rigorous research with conclusions that are appropriately drawn based on the data presented.

Reviewer #1: Yes

Reviewer #2: Yes

Reviewer #3: Yes

3. Has the statistical analysis been performed appropriately and rigorously?

Reviewer #1: Yes

Reviewer #2: Yes

Reviewer #3: Yes

4. Have the authors made all data underlying the findings in their manuscript fully available (please refer to the Data Availability Statement at the start of the manuscript PDF file)?

Reviewer #1: Yes

Reviewer #2: Yes

Reviewer #3: Yes

5. Is the manuscript presented in an intelligible fashion and written in standard English?

Reviewer #1: Yes

Reviewer #2: Yes

Reviewer #3: Yes

6. Review Comments to the Author

Reviewer #1: I see that the authors have addressed all my feedback and improved the piece.

Reviewer #2: I would like to thank the author for considering the given comments.

Reviewer #3: (No Response)

7. PLOS authors have the option to publish the peer review history of their article (what does this mean?). If published, this will include your full peer review and any attached files.

**Do you want your identity to be public for this peer review?** For information about this choice, including consent withdrawal, please see our Privacy Policy.

Reviewer #1: No

Reviewer #2: No

Reviewer #3: No
